# Universal and Expanded Screening Strategy for Congenital Cytomegalovirus Infection: Is Pool Testing by a Rapid Molecular Test in Saliva a New Choice in Developing Countries?

**DOI:** 10.3390/v16050772

**Published:** 2024-05-13

**Authors:** Giannina Izquierdo, Carolina Guerra, Roberto Reyes, Leslie Araya, Belén Sepulveda, Camila Cabrera, Pamela Medina, Eledier Mardones, Leonel Villavicencio, Luisa Montecinos, Felipe Tarque, William Acevedo, Marlon Barraza, Mauricio Farfán, Jocelyn Mendez, Juan Pablo Torres

**Affiliations:** 1Department of Pediatrics, Faculty of Medicine, University of Chile, Santiago 8820808, Chile; 2Neonatal Intensive Care Unit, Hospital Barros Luco Trudeau, Santiago, Chile; 3Centro de Investigación Clínica Avanzada (CICA), Hospital Exequiel González Cortés, Santiago, Chile; 4Molecular Biology Laboratory, Hospital Lucio Córdova, Santiago, Chile; 5Pharmacy Unit Santiago, Hospital Luis Calvo Mackenna, Santiago, Chile; mbarraza@calvomackenna.cl; 6Molecular Biology Laboratory, Hospital Luis Calvo Mackenna, Santiago, Chile; 7Division of Pediatric Infectious Diseases, Hospital Luis Calvo Mackenna, Santiago, Chile

**Keywords:** congenital cytomegalovirus, universal screening, expanded screening, pool testing, saliva

## Abstract

Background: Several screening strategies for identifying congenital CMV (cCMV) have been proposed; however, the optimal solution has yet to be determined. We aimed to determine the prevalence of cCMV by universal screening with saliva pool testing and to identify the clinical variables associated with a higher risk of cCMV to optimize an expanded screening strategy. Methods: We carried out a prospective universal cCMV screening (September/2022 to August/2023) of 2186 newborns, analyzing saliva samples in pools of five (Alethia-LAMP-CMV^®^) and then performed confirmatory urine CMV RT-PCR. Infants with risk factors (small for gestational age, failed hearing screening, HIV-exposed, born to immunosuppressed mothers, or <1000 g birth weight) underwent expanded screening. Multivariate analyses were used to assess the association with maternal/neonatal variables. Results: We identified 10 infants with cCMV (prevalence: 0.46%, 95% CI 0.22–0.84), with significantly higher rates (2.1%, 95% CI 0.58–5.3) in the high-risk group (*p* = 0.04). False positives occurred in 0.09% of cases. No significant differences in maternal/neonatal characteristics were observed, except for a higher prevalence among infants born to non-Chilean mothers (*p* = 0.034), notably those born to Haitian mothers (1.5%, 95% CI 0.31–4.34), who had higher odds of cCMV (OR 6.82, 95% CI 1.23–37.9, *p* = 0.04). Incorporating maternal nationality improved predictive accuracy (AUC: 0.65 to 0.83). Conclusions: For low-prevalence diseases such as cCMV, universal screening with pool testing in saliva represents an optimal and cost-effective approach to enhance diagnosis in asymptomatic patients. An expanded screening strategy considering maternal nationality could be beneficial in resource-limited settings.

## 1. Introduction

Cytomegalovirus (CMV) is the most common congenital infection and the leading cause of childhood hearing loss, cognitive deficits, and visual impairment, with an estimated prevalence of 0.2–5% worldwide [1,2,3,4]. Congenital CMV infection (cCMV) can occur after maternal primary infection (MPI) or non-primary maternal infection (MNPI) [5,6]. Only 10 to 15 percent of the infected infants are born with clinical, laboratory, or imaging manifestations such as intrauterine growth retardation, microcephaly, intracranial calcifications, jaundice, hepatosplenomegaly, thrombocytopenia, and petechiae [4]. Most infants with cCMV are asymptomatic at birth, but could develop long-term sequelae, including sensorineural hearing loss, which occurs in 5 to 15 percent of the children with asymptomatic cases during their first few years of life. 

Over the last few years, newborn screening for cCMV has become a more common strategy. The goal is early identification, treatment for symptomatic infants, and long-term follow-up to improve clinical outcomes. However, the optimal approach remains uncertain [7].

Universal screening programs test all newborns for cCMV in saliva, urine, or blood spots [8,9]. The saliva sample is easily accessible and does not cause the newborn discomfort, and also provides high sensitivity and specificity [10]. Cost, overuse of antiviral treatment, and potential parental distress are the main barriers to this strategy [11].

Targeted screening programs that include CMV testing of infants who fail their newborn hearing screening (NBHS) have become increasingly common. However, Fowler et al. found that only 57% of infants with cCMV-associated hearing loss in the neonatal period were identified by hearing-targeted cCMV screening protocols, which is suboptimal compared with universal screening [12,13]. 

Another screening alternative is expanded screening, which identifies factors in newborns who may be at higher risk for cCMV and those who do not pass the hearing test [14]. These include those born to HIV-infected or immunocompromised mothers, those who have a suspected maternal history of CMV infection, infants who are small for their gestational age (SGA), and/or those who have clinical or laboratory signs compatible with CMV infection [15,16]. The latter strategy allows us to expand the search for CMV to those with some suggestive criteria without significantly increasing costs. However, this would not allow us to identify the total number of cases of congenital CMV.

Some rapid molecular techniques have demonstrated appropriate sensitivity and specificity in detecting CMV in newborns [17]. Our research group recently performed a validation study of a point-of-care rapid molecular CMV test (Alethia-LAMP-CMV^®^ amplification assay) in saliva pools of five samples, which had a high concordance compared with the reference technique [18]. This could be a new and more cost-effective alternative to implementing universal cCMV screening due to the low prevalence of this infection and could be a more affordable approach in less developed regions with limited laboratory resources and detection capacity. 

This study aimed to determine the prevalence of cCMV by saliva pool testing as part of a universal screening strategy and to assess the factors associated with a higher risk of presenting with cCMV infection to optimize the factors included in an expanded screening strategy.

## 2. Materials and Methods

From September 2022 to August 2023, saliva swabs were prospectively collected with nylon-flocked swabs (Copan FLOQSwabs, Murrieta, CA, USA), 1 h after breastfeeding, from all preterms and term newborns less than 21 days of age born in the Maternity Ward of the Hospital Barros Luco, Santiago, Chile as part of a universal cCMV screening strategy. Enrollment occurred every week from Monday to Friday during working hours. Children referred from other hospitals after birth were excluded. The study was approved by the Ethics Committee of the Metropolitan South Health Service, Santiago, Chile, and informed consent was obtained from the mother or legal guardian prior to saliva sample collection.

We performed a rapid molecular test for CMV detection in saliva (Alethia-LAMP-CMV^®^ assay) [19] in pools of 5 samples. In positive pools, samples were tested individually with the Alethia-LAMP-CMV assay. All positive cases in saliva were confirmed by CMV RT-PCR in urine samples [4,20]. All newborns with cCMV risk factors (those who were SGA < p3 [14,15], infants who failed the NBHS [12], were HIV-exposed or were birthed by an immunosuppressed mother [21] and <1000 g of birth weight) [15] were evaluated in our center via urine CMV RT-PCR as part of an expanded screening strategy. This group was assessed using both techniques (saliva pool and urine RT-PCR) during the study period.

**Saliva sample.** To collect the specimen, a nylon-flocked swab was placed on the inner surface of both infants’ cheeks for 30 s or until the tip appeared saturated. Then, the swab was transferred into a dry, sterile tube. Saliva samples were processed at the Molecular Biology laboratory in the Hospital Lucio Córdova within the first 24 h after collection if they were stored at room temperature or within 7 days if they were stored at 2 to 8 °C [22]. Pool testing was performed as described and validated by Izquierdo et al. [16].

**Urine samples**. 

The sample was collected using pediatric urine collector bags, then placed in 50 mL flasks and stored at 4 °C until processing.

CMV molecular detection. 

For detection by the rapid molecular technique in saliva, we use the Alethia-LAMP-CMV^®^ assay, according to the manufacturer’s instructions [19]. This technique does not require prior extraction of nucleic acids from the sample. For CMV real-time PCR (RT-PCR), nucleic acid extraction was performed using MagDEA^®^ Dx SV [23] (automated extraction). To detect CMV on urine samples and quantify viral loads of CMV in whole blood, GeneProof Cytomegalovirus^®^ PCR Kit [24] was performed according to the manufacturer’s instructions.

We recorded the following variables in all mothers: age, parity, nationality, HIV status or other maternal immunodeficiency; and their screened newborns: gender, gestational age at birth, weight, birth weight for gestational age, such as SGA < p10, severe SGA < p3, and age at screening.


**Clinical assessment of newborns with confirmed cCMV infection.**


All newborns with a confirmed cCMV infection were evaluated at enrollment with (a) Complete physical examination; (b) a protocolized laboratory evaluation (hematological, biochemical, and microbiological studies were performed using standard techniques); (c) CMV viral loads in whole blood; (d) automated auditory brainstem response (BSER) before maternity ward discharge in all infected infants. All children were referred to the Otorhinolaryngology Department, and hearing was assessed with an auditory steady-state evoked potential; (e) ophthalmologic assessment at birth by fundoscopy and during the follow-up period; and (f) neuroimaging (US or MRI). Children were classified as mild, moderate–severe, and/or with CNS involvement and treated with antivirals according to the Chilean recommendation for cCMV infection [20]. Newborns were admitted to the Infectious Diseases Clinic of the Hospital Exequiel González Cortés for a long-term clinical follow-up (6 years). The study evaluated the relationship between cCMV and maternal and neonatal variables (see statistical analysis).


**Definitions**


Universal screening strategy: All newborns participating in the study underwent testing using a rapid molecular test on saliva pools, with positive cases confirmed through CMV RT-PCR on urine samples.

Expanded screening strategy: All newborns with one or more of the following risk factors were screened: severe SGA < p3, infants who failed the newborn hearing screening (NBHS), infants who were HIV-exposed or born to an immunosuppressed mother, and infants < 1000 g of birth weight; they then underwent CMV RT-PCR urine tests. Since they participated in the study, all these infants were also screened with pool testing of their saliva using the rapid molecular test according to the universal screening approach.

Symptomatic infection at birth: was defined as the presence of an abnormal physical examination (petechiae/ purpura, jaundice, hepatomegaly, splenomegaly, neurologic symptoms such as hypotonia, seizures, paresis, or weak sucking), chorioretinitis, small for gestational age (SGA), thrombocytopenia (platelet count < 100 × 10^3^/μL), elevated alanine aminotransferase levels (ALT > 80 IU/L), hyperbilirubinemia (direct bilirubin level > 2 mg/dL), microcephaly or neuroimaging abnormalities in cUS or MRI. SNHL was defined as a hearing threshold >25 dB tested by brainstem auditory evoked responses (BSER) in either ear [4,20]. Newborns who did not fulfill any of the criteria mentioned above after a complete evaluation at birth were considered to have asymptomatic cCMV.

SGA: was defined as a birth weight below the 10th percentile for gestational age, and severe SGA was defined as a birth weight below the 3rd percentile. Microcephaly was designated as an HC below an SD of −2 for gestational age. SNHL was defined as a hearing threshold >25 dB tested by BSER in either ear [4,20].

**Statistical analysis.** Bivariate and multivariate analyses, as well as predictive models, were used to assess CMV infection and the relationship between congenital cytomegalovirus and maternal and neonatal variables. Categorical variables were analyzed using the X2 test or Fisher’s test. For continuous variables, the Wilcoxon or Kruskal–Wallis nonparametric test was employed. Mean and proportion differences of various variables and their corresponding 95% confidence intervals (CI) were estimated. Bivariate logistic regression models were used to examine the association of variables. Variables with *p*-values below 0.1 were included in a multivariate logistic regression model. The strength of association was estimated by calculating crude and adjusted odds ratios (ORs) with their respective 95% CI. Multivariate logistic regression was used to construct predictive models for cCMV, which were evaluated using receiver operating characteristic (ROC) curves. Youden’s index estimated the cut-off point with the highest sensitivity and specificity. CMV prevalence and 95% CI were calculated for the study population and the high-risk group. Statistical analyses were performed using GraphPad Prism version 6.0 software (La Jolla, CA, USA) and RStudio version 4.2.3, considering a *p*-value of <0.05 as statistically significant.

## 3. Results

During the study period, 3171 infants were born at Barros Luco Trudeau Hospital, among whom 2198 (69%) were invited to participate. The parents of 12 infants refused enrollment (0.5%), 7 expressed disinterest in knowing whether their newborn was infected, while the remaining 5 were concerned about waiting to breastfeed. A total of 2186 neonates (99.5%), 52.6% males, underwent universal screening in 437 saliva pools employing the Alethia-LAMP-CMV^®^ rapid molecular technique. cCMV infection was confirmed by urine CMV RT-PCR prior to 21 days of life in 10/2186 neonates with a cCMV prevalence rate of 0.46% (95% CI 0.18–0.78) (Figure 1).

There were two false-positive results in the saliva pool test screening, representing 0.09% (2/2186) of cases. In both cases, the individual Alethia-LAMP-CMV^®^ test was positive for the individual sample, but the saliva CMV RT-PCR and the urine confirmatory test were negative. 

The mothers and birth characteristics of the total screened newborns are presented in Table 1. The monthly distribution of the ten cCMV cases diagnosed during the study period is shown in the Appendix A. The median age at diagnosis was 2 days (IQR 1–2).

All the cCMV-infected infants had a normal physical examination. The median platelet count, ALT, and direct bilirubin levels were normal. The median gamma-glutamyltransferase level in the blood (GGT) was 105 U/L (IQR 62.7–202.2), and the blood CMV viral load (VL) at diagnosis was 305 IU/mL (IQR 67.5–2802.5)/Log 2.6 (IQR 1.71–3.13). In an exploratory analysis, we found a significant correlation between the blood CMV VL and the GGT levels in the infected infants (r = 0.54; *p* = 0.01). (Appendix A).

Antiviral treatment was offered to 5 out of 10 infants. Two treatments were offered because of SNHL, one for sepsis-like syndrome, one for SGA < p3, and one for persistent high GGT values. One patient in the series died, but the death was not related to cCMV (*Klebsiella pneumoniae* bacteremia). Oral valganciclovir was the drug of choice in four cases, except in the infant with a sepsis-like syndrome, who received intravenous ganciclovir.

Of the total number of infants enrolled, 190 (8.7%) were newborns with any risk factor, who were involved in the cCMV study with urine RT-PCR CMV (expanded screening) in addition to the universal screening in saliva. Four infants with cCMV infection were found in this risk group, with a prevalence rate of 2.1% (95% CI 0.58–5.3), which was significantly higher than the average prevalence in newborns (*p* = 0.04). Only one infant in the risk group who was SGA and failed the NBHS presented a false-negative result for the saliva pool test. Additionally, Alethia-LAMP-CMV^®^ single and saliva RT-PCR were negative, but the urine RT-PCR and the blood viral load were positive (Case 10, Table 2).

To assess the risk factors for cCMV infection, we compared the group of newborns diagnosed with cCMV (*n* = 10) with the uninfected group (*n* = 2176) (Table 1). No significant differences were observed in maternal age, parity, HIV, or other immunosuppressed conditions in the mother. Moreover, there were no significant disparities in gestational age, median birth weight, or history of SGA. A higher prevalence of cCMV was found in infants born to non-Chilean mothers (70% vs. 39.2% in Chilean mothers (*p* = 0.034)). The cCMV prevalence according to the nationality of the mother was 1.5% (95% CI 0.31–4.34), 0.84% (95% CI 0.23–2.13), and 0.22% (95% CI 0.05–0.66) in Haitian, Venezuelan, and Chilean mothers, respectively. A significant difference was found only in Chilean and Haitian mothers *p* = 0.03. In the bivariate and multivariate analyses, newborns with a Haitian mother had an odds ratio of 5.59 (95% CI 0.93–32.8) and 6.82 (95% CI 1.23–37.9), respectively (*p* = 0.04) (Table 3). Conversely, in the bivariate analysis, normal birth weight had an odds ratio of 0.02 (95% CI 0.0–1.62) (*p* = 0.03) (Table 3).

We analyzed the predictive ability of the expanded strategy used in our study, including infants born to an HIV-positive or otherwise immunosuppressed mother, infants with an SGA < 1000 g, infants that failed the NBHS, and infants presenting with other clinical findings. A ROC curve was used to assess this prediction, and the model was then supplemented by including the significant clinical variables obtained from the bivariate and multivariate analysis of the present study, such as maternal nationality. The inclusion of these clinical variables improved the AUC of the ROC curve, from 0.65 (0.49–0.82) to 0.83 (0.71–0.95), as described in Figure 2.

## 4. Discussion

Developing an efficient, cost-effective, and minimally invasive screening strategy for cCMV is gaining increasing recognition as a significant public health issue. We conducted a prospective evaluation of 2186 infants using a universal screening strategy through pool testing in saliva; the cCMV prevalence rate was 0.46%. Mothers showed high acceptance of the saliva collection study, with only minimal refusal (0.5%) due to the safety of the procedure and a low number of false positives.

In addition, we identified several clinical variables that may optimize the extended screening strategy used in our patients. After we conducted bivariate and multivariate analyses of clinical variables to refine the definition of the risk group for cCMV, including factors such as the mother’s nationality, the accuracy of the extended screening strategy significantly improved. The ROC curve had an AUC of 0.83 (0.712–0.949), indicating a high level of accuracy. 

The universal screening strategy involved pool testing in saliva with a rapid molecular test, which was effective, as previously reported [18]. There were only two false-positive results and one false negative, showing that this technique has a strong correlation with CMV RT-PCR in urine. It should be noted that in 6 out of 10 (60%) of the detected cCMV cases, the newborns would not have undergone CMV testing under normal circumstances because they were asymptomatic at birth and did not belong to the high-risk group. It is crucial to highlight that asymptomatic patients may experience long-term sequelae [25]. Therefore, universal screening for cCMV in newborns allows for early detection and follow-up, which may improve clinical outcomes, particularly for asymptomatic infants. 

If a universal screening strategy is not feasible, an extended screening strategy may be a viable alternative, particularly if clinical variables that are relevant to the risk group for cCMV infection are included. 

Although limited published data are available on expanded screening strategies [14,26,27], our results are similar to those of a recently published study that evaluated an expanded testing protocol in Canada. Akiva et al. [14] conducted a retrospective analysis of 465 high-risk newborns tested for cCMV using an expanded screening strategy. The prevalence of cCMV infection was highest among infants tested due to primary maternal CMV infection (19%, 8/42), followed by those who failed initial NBHS (11.4%, 10/88), maternal HIV infection (2.2%, 3/137), and clinical suspicion alone (2.2%, 5/232). The authors suggest that these criteria should be considered additional criteria for expanded CMV screening, particularly in locations where universal screening is not yet the standard of care.

In a previous study, our research group prospectively examined 193 newborns. We identified high-risk groups with a high prevalence of cCMV, including those with clinical findings of congenital infection (1/33, 33.3%), infants exposed to HIV (2/41, 4.9%), SGA (3/90, 3.3%), and infants born with a birth weight < 1500 g (1/60, 1.7%). These criteria were used to define the expanded screening in our current study [16].

Suarez et al. [27] established expanded screening as a new alternative to cCMV screening to improve early detection rates. This approach includes a variety of known CMV-related symptoms, and thus offers an improved detection rate for cCMV cases compared to those tested under a hearing-targeted-only approach. However, this strategy primarily detects symptomatic cases compared to the universal screening strategy. On the other hand, the universal screening strategy in our study allowed for the detection of 6 out of 10 cCMV cases, all of which were clinically asymptomatic.

The bivariate and multivariate analysis of clinical variables in the enrolled children resulted in a proposal of additional variables for the risk group for cCMV, such as the mother’s nationality. Avika et al. and Suarez et al. did not consider maternal nationality as a risk factor for cCMV [14,27]. Nevertheless, there are reports of an increased prevalence of cCMV in children born to African American mothers, and the same was observed for Haitian mothers. Fowler et al. conducted a study on a cohort of approximately 100,000 newborns. They found that the population of African descent had a higher prevalence of cCMV, 0.95%, compared to 0.45% in the general population [28]. Our study found that infants born to Haitian mothers have a 6.8 times higher risk of developing cCMV. The prevalence of cCMV was 1.5% in children born to Haitian mothers, 0.84% in children born to Venezuelan mothers, and 0.22% in children born to Chilean mothers. Therefore, we suggest considering this variable as a risk factor for active screening; however, it is important to validate our results. None of the positive patients were infants born to HIV-infected mothers, who are known to be at increased risk for CMV. 

A comparison of the prevalence of cCMV in the general population and the high-risk group revealed a higher prevalence of 2.1% in the latter. Incorporating significant variables could potentially result in an even higher prevalence. Screening for cCMV based on risk factors could be considered an alternative to universal screening in resource-limited areas. 

One method of increasing testing efficiency and conserving resources in a universal screening strategy is pooling saliva samples. Saliva is advantageous for screening purposes because it is easy and quick to collect, especially in premature infants. It has a high sensitivity and a predictive negative value. However, if a saliva sample tests positive for CMV, it should be confirmed with a CMV PCR test on a urine specimen, which is considered to be the gold standard. This is necessary because false-positive results can occur due to recent breastfeeding [5]. During the 13-month study period, 15,805 infants (93.6% of all live newborn infants) in Israel were screened for cCMV using the pooled approach, with a birth prevalence of 0.34% [29]. This study and others demonstrate the wide feasibility and benefits of pooled saliva testing as an efficient, cost-saving, and sensitive approach to universal screening of cCMV with CMV RT-PCR [30,31,32].

Our study used a novel point-of-care platform for cCMV diagnosis in saliva, providing screening programs in centers that lack access to molecular biology laboratories. This is particularly beneficial in low-income countries, where it will facilitate universal screening [18].

In a preliminary analysis, a significant correlation between GGT levels and VL in the blood of children with cCMV was found in our study. This finding suggests that GGT levels could be an additional biomarker for evaluating newborns and defining antiviral treatment. Validation studies in a larger group of subjects must confirm this potential biomarker. The low prevalence of cCMV in our series (0.46%) is noteworthy compared to the 1.8% reported in our country in the 1990s by Luchsinger et al. [33]. This difference may be attributed to the control measures implemented during the SARS-CoV-2 pandemic, which were lifted in our country by October 2022, as well as recent socio-cultural, economic, and developmental changes. Other international reports have also described decreased cCMV prevalence in pandemic contexts in which cCMV screening is conducted [34].

Our study has several limitations. The study was conducted in one of the largest maternity hospitals in the country, which delivers 3900 newborns annually. However, it should be noted that the findings may not represent the entire national population. Data were collected prospectively from 69% of all newborns in the hospital on weekdays. Enrolling participants throughout the weekend could enhance the external validity of the findings. Validation is necessary before implementing the expanded screening strategy, which includes the mother’s nationality. Local validation of risk factors may be required to develop the risk factors included in this strategy, as there may be multiple factors depending on local conditions. We did not conduct an economic or cost–benefit evaluation of the different screening strategies. We estimate that pool testing in saliva can reduce costs by a factor of four [16]. However, we did not calculate the comparative costs of universal and extended screening, so further research on this is needed. Finally, Avika et al. identified maternal diagnosis of primary infection as a potential risk factor for cCMV [12]. However, we did not explore this because, in developing countries such as ours, screening for infection during pregnancy is not routinely performed. Due to the high seroprevalence of CMV in pregnant women in our countries, universal neonatal screening may be more appropriate given the complex interpretation and availability of maternal serology in this setting of high maternal prevalence.

Some of the challenges our research may face in the future are as follows: to validate pool testing in saliva with a larger number of samples, which could make it more cost-effective; to perform a randomized comparative study between the universal screening strategy and extended screening for cCMV screening, especially in developing countries where the universal screening strategy can be costly. Finally, we aim to expand the study to a national or regional level to strengthen the external validity of our results.

## 5. Conclusions

In conclusion, for low-prevalence diseases such as cCMV, universal screening with pool testing represents an optimal and cost-effective approach to enhance diagnosis in asymptomatic patients. Alternatively, if financial resources and facilities are unavailable to implement universal screening, an expanded screening strategy incorporating new clinical variables, such as the mother’s nationality, may be a suitable option. However, it is important to note that this strategy does not detect all cases of cCMV. Universal screening by pool testing in saliva or an expanded strategy may be useful in regions with limited diagnostic capacity.

## Figures and Tables

**Figure 1 viruses-16-00772-f001:**
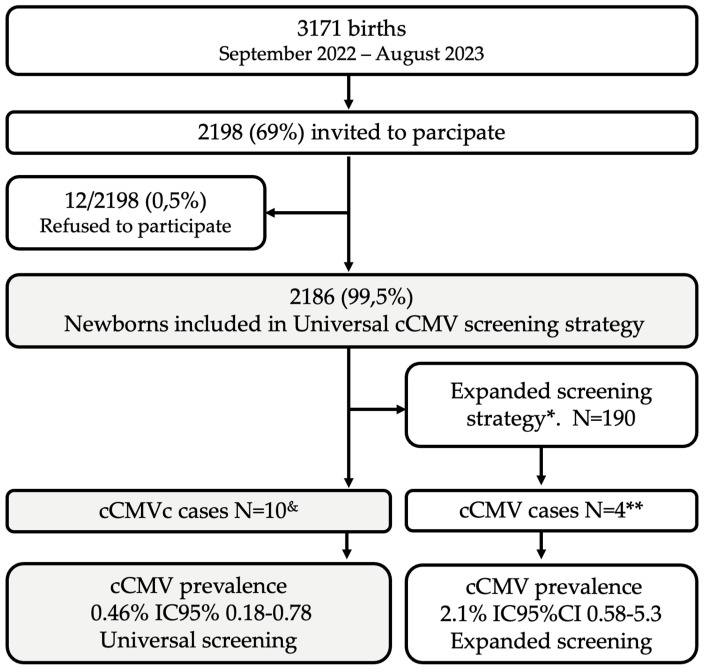
Flowchart for cCMV screening using different screening strategies. * Expanded screening strategy: CMV urine RT-PCR in all small-for-gestational-age (SGA), infants who failed hearing screening, infants who were HIV-exposed or born from an immunosuppressed mother, and <1000 g. ** cCMV cases were detected by using both expanded and universal screening strategies. Only ^&^ one case in ten was detected by an expanded screening with urine CMV PCR.

**Figure 2 viruses-16-00772-f002:**
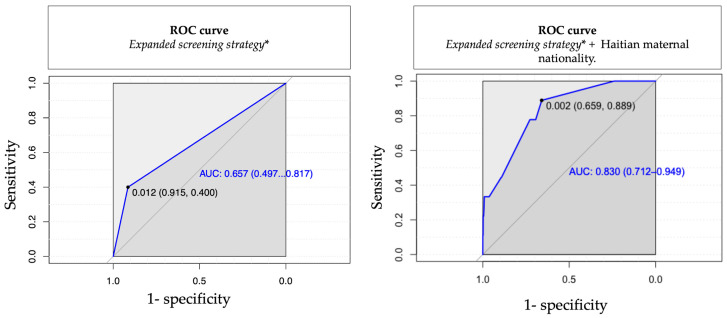
Predictive models for cCMV screening strategies. * Expanded screening strategy: CMV urine RT-PCR in all small-for-gestational-age (SGA) infants, infants who failed hearing screening, infants who were HIV-exposed or born to an immunosuppressed mother, and infants < 1000 g.

**Table 1 viruses-16-00772-t001:** Characteristics of mothers and newborns screened for congenital cytomegalovirus (cCMV) infection by pool testing in saliva.

	Total N = 2186	cCMV (−) Newborns N = 2176	cCMV (+) Newborns N = 10	*p*
**Mothers**				
**Age** (median/IQR)	29 (25–34)	29 (25–34)	29 (25–34)	*0.74*
**Nacionality n (%)** - Chilean - Venezuelan - Haitian - Others	1325 (60.6) 473 (21.6) 196 (9) 192 (8.8)	1322 (60.8) 469 (21.5) 193 (8.9) 192 (8.8)	3 (30) 4 (40) 3 (30) 0 (0)	* **0.034 *** *
**Parity n (%)** - Nuliparous - Multiparous	2070 797 (38.5) 1273 (61.5)	2060 793 (38.5) 1267 (61.5)	10 4 (40) 6 (60)	*0.81*
**Newborns risk factors of cCMV**				
**Gestational age (weeks)** (median/IQR)	38 (37–40)	38 (37–41)	38.5 (33.5–39)	*0.48*
**Birth weigth (g)** (median/IQR)	3212 (2795–3506)	3214 (2800–3568)	2745 (2205–3326)	*0.09*
**HIV status n (%)** - Positive - Negative	10 (0.5) 2176 (99.5)	10 (0.5) 2166 (99.5)	0 10 (100)	*1*
**SGA < p10 ** n (%)**	345 (15.8)	341 (15.7)	4 (40) ***	*0.08*

* *p* value < 0.05/ ** SGA: small for gestational age/ *** SGA < p3.

**Table 2 viruses-16-00772-t002:** cCMV newborns diagnosed by universal and expanded screening with pool testing in saliva.

	Alethia-LAMP-CMV pool	Individual Alethia-LAMP-CMV Assay	Saliva CMV RT-PCR	Viral Load in Saliva (Copies/mL)	Urine CMV RT-PCR *	Viral Load in Blood	Maternal Nationality	*Screening* *Strategy*
**Case 1**	Positive	Positive	Positive	19,100 Log 4.28	Positive	7160 Log 3.85	Chilean	Universal
**Case 2**	Positive	Positive	Positive	5800 Log 3.76	Positive	66,800 Log 4.82	Haitian	Universal and Expanded (SGA < p3)
**Case 3**	Positive	Positive	Positive	456,500 Log 5.65	Positive	<35 Log 1.54	Venezuelan	Universal
**Case 4**	Positive	Positive	Positive	705,000 Log 5.84	Positive	1260 Log 3.01	Venezuelan	Universal and Expanded (<1000 g)
**Case 5**	Positive	Positive	Positive	3,240,000 Log 6.51	Positive	73 Log 1.86	Venezuelan	Universal
**Case 6**	Positive	Positive	Positive	71,500,000 Log 7.85	Positive	1350 Log 3.13	Venezuelan	Universal
**Case 7**	Positive	Positive	Positive	24,300,000 Log 7.38	Positive	405 Log 2.61	Chilean	Universal
**Case 8**	Positive	Positive	Positive	11,300,000 Log 7.05	Positive	51 Log 1.71	Chilean	Universal and Expanded (SGA < p3)
**Case 9**	Positive	Positive	Positive	Inhibited RT-PCR	Positive	206 Log 3.1	Haitian	Universal
**Case 10**	Negative	Negative	Negative	Negative	Positive	80 Log 1.9	Haitian	Universal and Expanded (SGA < p3 and failed NBHS)

Expanded screening: CMV urine RT-PCR in all small for gestational age (SGA), infants who failed newborn hearing screening (NBHS), HIV-exposed or born from an immunosuppressed mother and infants < 1000 g). * Gold standard technique for cCMV diagnosis.

**Table 3 viruses-16-00772-t003:** Bivariate and multivariate analyses of the maternal and newborn variables of cCMV and uninfected cases.

	Bivariate Analyses			Multivariate Analyses	
Variable	OR	CI 95%	*p*	OR	CI 95%
**Maternal nationality** - Chilean - Venezuelan - Haitian - Other	3.96 5.59 ND	0.81–22.1 0.93–32.8 ND	0.08 0.04 * 0.99	4.22 6.82 0.0	0.92–21.7 1.23–37.9 * ND
**Birth weight (g)** - <1000 - <1500 - <2500 - <4000	1.92 0.12 0.02	0.02–81.1 0.0–10 0.0–1.62	0.73 0.27 0.035 *	0.43 0.31 0.07	0.02–11.6 0.04–6.6 0.01–1.45

ND: no data/* Statistical significance.

## Data Availability

The data from this study can be made available upon request to Dr. Giannina Izquierdo (gizquierdo@uchile.cl).

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
