# Peer review of "Universal and Expanded Screening Strategy for Congenital Cytomegalovirus Infection: Is Pool Testing by a Rapid Molecular Test in Saliva a New Choice in Developing Countries?"

_viruses, 2024, doi:10.3390/v16050772_

Round 1
Reviewer 1 Report
Comments and Suggestions for Authors
This is clearly a worthwhile study but I am not clear whether it represents …
1. a novel and scientifically rigorous assessment of the use of saliva pools…or …
2. a description of the prevalence of cCMV in your population as determined by a manageable test regime. This would not be internationally novel.
For #1, the title is perfect. However the outcome of individual testing high risk individuals is largely irrelevant….as are the demographic factors (eg: maternal nationality). Rather the paper needs to focus on the creation and testing of saliva pools from low-risk individuals. It should then answer questions like
a. why choose 5 samples/per pool?...why not 10?
b. Are positives missed by creation of pools?
For #2, the title needs to change and the conclusion stated in the abstract (“Universal saliva pool testing enhanced cCMV detection by 60%.”) is hard to understand…of course more tests detect more positives…
I also have a problem with Table 1 as it includes the high risk babies …such as those with HIV.
Do Haitians have a higher incidence of HIV positivity or socio-economic metrics?
Table 2 is not needed and I cant follow Table 3. I also don't think Fig 3 adds anything.
Overall the paper should be shortened ....
Comments on the Quality of English Language
The English is mostly OK
The use of italics and underlining should be avoided
Author Response
Thank you very much for taking the time to review this manuscript. Please find the detailed responses below (in blue) and the corresponding revisions/corrections highlighted/in track changes in the re-submitted files.
Reviewer 1
This is clearly a worthwhile study but I am not clear whether it represents …
- a novel and scientifically rigorous assessment of the use of saliva pools…or …
- a description of the prevalence of cCMV in your population as determined by a manageable test regime. This would not be internationally novel.
For #1, the title is perfect. However the outcome of individual testing high risk individuals is largely irrelevant….as are the demographic factors (eg: maternal nationality). Rather the paper needs to focus on the creation and testing of saliva pools from low-risk individuals.
We appreciate the reviewer's comment, which we believe is correct and appropriate. We have made changes in the manuscript, both in the introduction and in the discussion, aiming at the approach suggested by the reviewer to focus on the creation and testing of saliva pools from low-risk individuals.
It should then answer questions like
- why choose 5 samples/per pool?...why not 10?
Pool sizes were determined based on our previous experience, where we were able to validate the study by pool testing for the detection of SARS-CoV-2 (reference: Optimizing RT-PCR detection of SARS-CoV-2 for developing countries using pool testing. Rev Chilena Infectol. 2020 Jun;37(3):276-280).
This decision considered a pool size that would be manageable for laboratory personnel. When it is necessary to "open" a pool, if the pool is very large, it can be more complex to analyze in the laboratory, could be associated with more errors and eventually a longer turnaround time. In addition, we were able to confirm that a pool size of 5 samples maintains a very adequate NPA and PPA values.
Izquierdo, G., Farfan, M.J., Villavicencio, L. et al. Optimizing congenital cytomegalovirus detection by pool testing in saliva by a rapid molecular test. Eur J Pediatr 182, 5131–5136 (2023). https://doi.org/10.1007/s00431-023-05183-x
We agree with the reviewer's comment that the next step should be to advance the validation of pool testing with 10. This could make it even more attractive to have universal screening strategies for cCMV, a low-prevalence disease
- Are positives missed by creation of pools?
In our experience (previously published), no loss of positives is observed when analyzing samples in a pool of 5. In the present study, we had only 2 false positive samples and only 1 false negative sample out of a total of 2186 infants screened. Considering these results, we can estimate that the missed positives are marginal, being consistent with the possibility of screening for cCMV using pool testing in saliva by a rapid molecular method.
Izquierdo, G., Farfan, M.J., Villavicencio, L. et al. Optimizing congenital cytomegalovirus detection by pool testing in saliva by a rapid molecular test. Eur J Pediatr 182, 5131–5136 (2023). https://doi.org/10.1007/s00431-023-05183-x
For #2, the title needs to change and the conclusion stated in the abstract (“Universal saliva pool testing enhanced cCMV detection by 60%.”) is hard to understand…of course more tests detect more positives…
We agree with the reviewer and have revised the title and abstract to better align with their recommendations. We changed the title and eliminated the phrase:”universal saliva pool testing enhanced cCMV detection by 60%”.
Conclusions: For low-prevalence diseases such as cCMV, universal screening with pool testing in saliva represents an optimal and cost-effective approach to enhancer diagnosis in asymptomatic patients. In resource-limited settings, an expanded screening strategy considering maternal nationality could be beneficial.
I also have a problem with Table 1 as it includes the high risk babies …such as those with HIV.
In response to the reviewer's suggestion, Table 1 has been modified to include all risk group newborns, including those born to a mother with HIV, small for gestational age (SGA), and preterm infants and low birth weight. This modification is intended to enhance the clarity and understanding of Table 1.
In the entire group, there are 10 newborns born to a positive HIV mother, but none of them are HIV infected.
Do Haitians have a higher incidence of HIV positivity or socio-economic metrics?
Since 2017, Chile has become a popular destination for migrants from South America and the Caribbean region (low- and middle-income countries migration). Fuster et al. conducted a survey of selected transmissible conditions. 498 Haitian participants (60.4% female) from 10 communities in two regions of Chile were interviewed. They found a higher prevalence of HIV 2.4% (95 CI 1.3-4.2%) and hepatitis B (HBsAg positive) 3.4% (95 CI 2.1-5.5%) than in the Chilean population.
However, none of the infants with cCMV infection were born to HIV-positiveHaitian mothers.
The higher prevalence of cCMV in our study could be due to African-American race (higher prevalence) or socio-cultural factors that may predispose to reactivation or reinfection, which should be evaluated in a future study.
Fuster, F., Peirano, F., Vargas, J.I. et al. Infectious and non-infectious diseases burden among Haitian immigrants in Chile: a cross-sectional study. Sci Rep 10, 22275 (2020). https://doi.org/10.1038/s41598-020-78970-3
Table 2 is not needed and I cant follow Table 3. I also don't think Fig 3 adds anything.
Table 2
We believe that Table 2 facilitates a better understanding of the data found in our study and shows the results of the different variables and conditions of the infants with cCMV. Therefore, we would like to keep Table 2 in the manuscript.
Table 3.
In order to facilitate the interpretation of Table 3, the title has been modified in accordance with the recommendations of Reviewer 1.
Figure 3
As proposed by Reviewer 1 and perceived as a source of confusion by Reviewer 2, we have removed Figure 3.
Overall the paper should be shortened ....
Given the reviewers' comments, we had to add certain additional clarifications to the text of the manuscript, which is within the number of words accepted by the journal for this type of article.

Reviewer 2 Report
Comments and Suggestions for Authors
In this study universal and expanded screening protocols to detect cCMV in newborns were evaluated. For the universal screening pooled saliva samples were tested by a lamp method. With this screening 10/2186 newborns with cCMV were detected and only one child with cCMV was missed (only urine positive for CMV). With the expanded screening of only 190 newborns six newborns with cCMV were not diagnosed. The advantage of universal screenings was already shown in several manuscripts. The authors should emphasize their special additional findings in the discussion.
Some minor amendments:
Please comment on the use of saliva samples instead of urine samples or dried blood spots.
Is it possible to speculate on the higher prevalence of cCMV in Haitian mothers?
In Figure 1 ten cCMV cases by universal screening are noted, however, in Table 2 only nine cases are shown (case 10 is the false negative one). Please clarify.
Table 1, last column p, second line 0,034*: Where is the explanation for the asterix?
Table 1, column three, last line 4 (40)**: Where is the explanation for the asterix?
Page 5, line 216-217: the odds ratio of 6.82 is from the multivariate analysis, in the bivariate analysis the odds ratio 5.59. Please clarify.
Figure 3: Two of the four cases of cCMV found by “expanded screening” are not in the circle “universal screening”. Please clarify.
Author Response
Reviewer 2
Thank you very much for taking the time to review this manuscript. Please find the detailed responses below (in blue) and the corresponding revisions/corrections highlighted/in track changes in the re-submitted files
In this study universal and expanded screening protocols to detect cCMV in newborns were evaluated. For the universal screening pooled saliva samples were tested by a lamp method. With this screening 10/2186 newborns with cCMV were detected and only one child with cCMV was missed (only urine positive for CMV). With the expanded screening of only 190 newborns six newborns with cCMV were not diagnosed. The advantage of universal screenings was already shown in several manuscripts. The authors should emphasize their special additional findings in the discussion.
We appreciate your comments and, in accordance with the reviewer's suggestions, we have reinforced the message of the use of universal screening for cCMV through a new technique in saliva, which has been demonstrated to have good sensitivity and specificity, feasibility, and a lower cost.
Some minor amendments:
Please comment on the use of saliva samples instead of urine samples or dried blood spots.
Saliva collection has high parental acceptance and is easier to collect than urine with high sensitivity (93-100%) and predictive negative value (98-99%), however a positive CMV PCR result on a saliva should be confirmed with a CMV PCR on a urine specimen (gold standard) due to false positive results as a consequence of recent breastfeeding.
Furthermore, urine specimens may be difficult to collect in preterm infants due to skin conditions and low urine output in the first days of life.
In our reality, dried blood spots are not always available and have different sensitivities reported in the literature.
Dollard SC, Dreon M, Hernandez-Alvarado N, Amin MM, Wong P, Lanzieri TM, Osterholm EA, Sidebottom A, Rosendahl S, McCann MT, Schleiss MR. Sensitivity of Dried Blood Spot Testing for Detection of Congenital Cytomegalovirus Infection. JAMA Pediatr. 2021 Mar 1;175(3):e205441. doi: 10.1001/jamapediatrics.2020.5441. Epub 2021 Mar 1. PMID: 33523119; PMCID: PMC7851756.
We added the point in the page 10 of the discussion as suggested by reviewer 2:
Saliva is advantageous for screening purposes due to its ease and quick collection compared to urine bags, especially in premature infants; with high sensitivity and predictive negative value, however a positive CMV result on a saliva should be confirmed with a CMV PCR on a urine specimen (gold standard) due to false positive results as a consequence of recent breastfeeding[ref B].
Is it possible to speculate on the higher prevalence of cCMV in Haitian mothers?
As we stated in the discussion, Fowler et al. conducted a study on a cohort of approximately 100,000 newborns and found that the population of African descent had a higher prevalence of cCMV, 0.95% compared to 0.45% in the general population [Fowler 2018; 25].
There is no data on CMV seroprevalence in Haitian women. A systematic literature review of the global seroprevalence of cytomegalovirus in Latin American countries showed a higher seroprevalence in pregnant women compared with EEUU and Europe (Fowler 2022). Lanzieri et al. also shows a higher prevalence of congenital CMV infection in countries with higher maternal seroprevalence like LAtin American countries.
In our study, the risk of developing cCMV was found to be 6.8 times higher in newborns born to Haitian mothers. The prevalence of cCMV was 1.5% in children born to Haitian mothers, 0.84% in children born to Venezuelan mothers, and 0.22% in children born to Chilean mothers. Therefore, we suggest considering this variable as a risk factor for active screening, but it is important to validate our results.
We hypothesize that Haitian mothers, being of African descent, may have a higher seroprevalence of CMV and therefore a higher risk of transmission to their newborns. Non-primary infection (reinfection or reactivation) accounts for the vast majority of neonatal infections in underdeveloped countries.
None of the positive patients were children of HIV-infected mothers, who are known to be at increased risk for CMV.
In Figure 1 ten cCMV cases by universal screening are noted, however, in Table 2 only nine cases are shown (case 10 is the false negative one). Please clarify.
The reviewer is correct in his/her observation. Indeed, Figure 1 shows 10 cases of cCMV and Table 2 shows 9 cases of cCMV plus another case that was confirmed as cCMV but was a false negative in the universal screening with pool testing in saliva, which was only confirmed by PCR in urine since the infant corresponded to the risk group. To clarify this point, we have added a brief note in Figure 1 and Table 2 to facilitate a better understanding of the data described.
Table 1, last column p, second line 0,034*: Where is the explanation for the asterix?:
A higher prevalence of cCMV was found in infants born to non-Chilean mothers (70% vs. 39,2% ) in Chilean mothers (p=0.034)). The reviewer is correct, we corrected the p-value asterisk and added it to the table.
Table 1, column three, last line 4 (40)**: Where is the explanation for the asterix?
The 4 patients of the cCMV group where SGA but less than p3
Page 5, line 216-217: the odds ratio of 6.82 is from the multivariate analysis, in the bivariate analysis the odds ratio 5.59. Please clarify.
In fact, the reviewer is right, we did not include the OR value of the bivariate analysis. it has been corrected in the manuscript.
“In the bivariate and multivariate analyses, newborns with a Haitian mother had an odds ratio of 5.59 (95% CI 0.93 – 32.8) and 6.82 (95%CI 1.23-37.9) respectively (p=0.04) (Table 3).”
Figure 3: Two of the four cases of cCMV found by “expanded screening” are not in the circle “universal screening”. Please clarify.
As suggested by Reviewer 1, and which was a source of confusion for Reviewer 2, we have removed figure number 3.

Reviewer 3 Report
Comments and Suggestions for Authors
I would like to thank the authors for their submission and allowing me to review their work.
This is an interesting study on an important topic. However, I would be grateful if you could add further explanations and changes on the following points:
1) INTRODUCTION: Page 2, lines 49, 53, 58, 64, and so on…
The period should be placed after the bracket.
2) INTRODUCTION
In the introduction section, I suggest specifying the difference between “primary infection” (which occurs when an individual without immunity against CMV becomes infected for the first time), and non-primary infection (which is due to reinfection with exogenous CMV strains or reactivation of latent endogenous CMV). This distinction is very important because it implies different management strategies [I suggest citing the following article: Congenital Cytomegalovirus and Hearing Loss: The State of the Art. J Clin Med. 2023;12(13):4465. doi:10.3390/jcm12134465].
3) MATERIALS AND METHODS: Page 3, Line 123
What audiological evaluations were performed?
4) MATERIALS AND METHODS: Page 3, Line 128
Were children with cCMV infection evaluated to exclude vestibular disorders?
5) MATERIALS AND METHODS: Page 3, Line 128
Were children with cCMV infection also tested for genetic mutations?
6) MATERIALS AND METHODS: Page 3, Line 140
Symptomatic cCMV infections are usually distinguished into “mild”, “moderate” or “severe” according to the severity of the signs and symptoms. What classification did you use to categorize symptomatic children? (REFENCES: Congenital Cytomegalovirus: A European Expert Consensus Statement on Diagnosis and Management. Pediatr Infect Dis J. 2017;36(12):1205-1213. doi:10.1097/INF.0000000000001763; Congenital cytomegalovirus infection in pregnancy and the neonate: consensus recommendations for prevention, diagnosis, and therapy. Lancet Infect Dis. 2017;17(6):e177-e188. doi:10.1016/S1473-3099(17)30143-3).
7) MATERIALS AND METHODS: Page 3, Line 150
What classification of hearing loss was used? How many children were diagnosed with SNHL?
8) RESULTS: Page 4, Line 176
I suggest specifying the gender of the study population.
9) DISCUSSION: Page 10, Line 388
I suggest adding the important concept that universal screening for cCMV is of paramount importance to identify asymptomatic children who may not only develop hearing loss, but also motor delay, speech-language delay, cognitive delay, and balance disorders over time. The prevalence of adverse audiological and neurological outcomes is indeed nonnegligible even among asymptomatic children with cCMV [I suggest citing the following article: Hearing outcomes in children with congenital cytomegalovirus infection: From management controversies to lack of parents' knowledge. Int J Pediatr Otorhinolaryngol. 2023;164:111420. doi:10.1016/j.ijporl.2022.111420]
10) DISCUSSION: Page 10, Line 388
I suggest adding the future prospects of this study.
Comments on the Quality of English LanguageMinor editing required
Author Response
Thank you very much for taking the time to review this manuscript. Please find the detailed responses below (in blue) and the corresponding revisions/corrections highlighted/in track changes in the re-submitted files
I would like to thank the authors for their submission and allowing me to review their work.
This is an interesting study on an important topic. However, I would be grateful if you could add further explanations and changes on the following points:
1) INTRODUCTION: Page 2, lines 49, 53, 58, 64, and so on…
The period should be placed after the bracket.
We changed the point to after the bracket according to the reviewer's suggestions.
2) INTRODUCTION
In the introduction section, I suggest specifying the difference between “primary infection” (which occurs when an individual without immunity against CMV becomes infected for the first time), and non-primary infection (which is due to reinfection with exogenous CMV strains or reactivation of latent endogenous CMV). This distinction is very important because it implies different management strategies [I suggest citing the following article: Congenital Cytomegalovirus and Hearing Loss: The State of the Art. J Clin Med. 2023;12(13):4465. doi:10.3390/jcm12134465].
We have added the concept of primary and non-primary infection as suggested by the reviewer. We have also included updated references on this topic.
A Aldè M, Binda S, Primache V, Pellegrinelli L, Pariani E, Pregliasco F, Di Berardino F, Cantarella G, Ambrosetti U. Congenital Cytomegalovirus and Hearing Loss: The State of the Art. J Clin Med. 2023 Jul 3;12(13):4465. doi: 10.3390/jcm12134465. PMID: 37445500; PMCID: PMC10342520.
B Leruez-Ville M, Chatzakis C, Lilleri D, Blazquez-Gamero D, Alarcon A, Bourgon N, Foulon I, Fourgeaud J, Gonce A, Jones CE, Klapper P, Krom A, Lazzarotto T, Lyall H, Paixao P, Papaevangelou V, Puchhammer E, Sourvinos G, Vallely P, Ville Y, Vossen A. Consensus recommendation for prenatal, neonatal and postnatal management of congenital cytomegalovirus infection from the European congenital infection initiative (ECCI). Lancet Reg Health Eur. 2024 Apr 1;40:100892. doi: 10.1016/j.lanepe.2024.100892. PMID: 38590940; PMCID: PMC10999471.
3) MATERIALS AND METHODS: Page 3, Line 123
What audiological evaluations were performed?
Hearing evaluation: automated brainstem auditory evoked responses (BSER) before maternity ward discharge in all infected infants. All children were referred to the Otorhinolaryngologist Department and hearing evaluation was performed with an auditory steady-state evoked potential
Thank you for your comment, we have corrected it in the text.
4) MATERIALS AND METHODS: Page 3, Line 128
Were children with cCMV infection evaluated to exclude vestibular disorders?
We know that vestibular problems are as prevalent (or more prevalent) than SNHL, but we also know that they are difficult to assess early in life. We strongly agree with the reviewer, and we should move forward with this evaluation in the infant period (6-8 months).
To date, we have not assessed ventricular function in infants with congenital CMV.
5) MATERIALS AND METHODS: Page 3, Line 128
Were children with cCMV infection also tested for genetic mutations?
No, unfortunately we don't have the ability to study them genetically. It could be a future collaborative study.
6) MATERIALS AND METHODS: Page 3, Line 140
Symptomatic cCMV infections are usually distinguished into “mild”, “moderate” or “severe” according to the severity of the signs and symptoms. What classification did you use to categorize symptomatic children? (REFERENCES: Congenital Cytomegalovirus: A European Expert Consensus Statement on Diagnosis and Management. Pediatr Infect Dis J. 2017;36(12):1205-1213. doi:10.1097/INF.0000000000001763; Congenital cytomegalovirus infection in pregnancy and the neonate: consensus recommendations for prevention, diagnosis, and therapy. Lancet Infect Dis. 2017;17(6):e177-e188. doi:10.1016/S1473-3099(17)30143-3).
We used the national guideline classification of cCMV (Izquierdo et al), which is very close to the 2017 European Consensus (Luck et al). We distinguished between mild, moderate, severe and children with central nervous system involvement (which includes SHN) to classify children and to decide when to treat them.
“Children were classified and treated with antivirals according to the Chilean Recommendation for cCMV infection “
Izquierdo G, Sandoval A, Abarzua F, Yamamoto M, Rodriguez JG, Silva M, et al. (2021) Recommendations for the diagnosis and management of cytomegalovirus infection in pregnant women and newborn infants. Rev Chilena Infectol 38(6):824-56. doi: 10.4067/s0716-10182021000600824.
7) MATERIALS AND METHODS: Page 3, Line 150
What classification of hearing loss was used? How many children were diagnosed with SNHL?
SNHL was defined as a hearing threshold >25 dB tested by brainstem auditory evoked responses (BSER) in either ear. We added the definition in the Materials and Methods section. Page 4.
Two patients failed the hearing screening at birth and initiated treatment with valganciclovir. BSER were normal at one and three months of age, and at the six-month follow-up none required support with hearing aids or cochlear implants.
8) RESULTS: Page 4, Line 176
I suggest specifying the gender of the study population. 52,6% males.
We added the information in the results. Page 4.
9) DISCUSSION: Page 10, Line 388
I suggest adding the important concept that universal screening for cCMV is of paramount importance to identify asymptomatic children who may not only develop hearing loss, but also motor delay, speech-language delay, cognitive delay, and balance disorders over time. The prevalence of adverse audiological and neurological outcomes is indeed nonnegligible even among asymptomatic children with cCMV [I suggest citing the following article: Hearing outcomes in children with congenital cytomegalovirus infection: From management controversies to lack of parents' knowledge. Int J Pediatr Otorhinolaryngol. 2023;164:111420. doi:10.1016/j.ijporl.2022.111420]
We agree with reviewer 3 and have added the suggestion and reference to the discussion.
“The universal screening strategy using pool testing in saliva with a rapid molecular test performed adequately, as previously reported [16]. There were only two false positives results and one false negative, showing a strong correlation of this technique with CMV RT-PCR in urine. It should be noted that 60% of the cases of cCMV detected, 6 out of 10 newborns, would not have undergone CMV testing under normal circumstances because they are asymptomatic at birth and not belong to the high-risk group. It is essential to emphasize that asymptomatic patients may experience long-term sequelae [C]. Therefore, newborn universal screening for cCMV allows for early detection and follow up, which may improve clinical outcomes, especially in the asymptomatic infants.”
10) DISCUSSION: Page 10, Line 388
I suggest adding the future prospects of this study.
Among the future prospects of our study, we can mention certain challenges: to validate pool testing in saliva with a larger number of samples, which could make it more cost-effective; to perform a randomized comparative study between the universal screening strategy and extended screening for cCMV screening, especially in developing countries where the universal screening strategy can be costly. Finally, we aim to extend the study to a national or regional level to strengthen the external validity of our results.
We added these points to the discussion.

Round 2
Reviewer 1 Report
Comments and Suggestions for Authors
The paper has been improved by the revisions
Comments on the Quality of English LanguageNo major problems